# Developmental Endothelial Locus-1 Promotes Osteoclast Differentiation and Activation

**DOI:** 10.3390/ijms26062673

**Published:** 2025-03-16

**Authors:** Kentaro Imamura, Keita Tachi, Tadahiro Takayama, Hironori Kasai, Ryutaro Shohara, Kenji Inoue, Yoichiro Taguchi, Saki Nakane-Koyachi, Atsushi Saito, Seiichi Yamano

**Affiliations:** 1Department of Periodontology, Tokyo Dental College, Chiyoda-ku, Tokyo 101-0061, Japan; imamurakentarou@tdc.ac.jp (K.I.); nakanesaki1013@gmail.com (S.N.-K.); atsaito@tdc.ac.jp (A.S.); 2Oral Health Science Center, Tokyo Dental College, Chiyoda-ku, Tokyo 101-0061, Japan; 3Komazawa Parkside Dental Clinic, Setagaya-ku, Tokyo 154-0021, Japan; tachi-k@dent.showa-u.ac.jp; 4Department of Periodontology, Nihon University School of Dentistry, Chiyoda-ku, Tokyo 101-0062, Japan; takayama.tadahiro@nihon-u.ac.jp; 5Kasai Dental Clinic, Kitakyushu-shi 800-0226, Fukuoka, Japan; kasaipirorikin@gmail.com; 6Shohara Dental Clinic, Osaka-shi 543-0013, Osaka, Japan; r.shohara@me.com; 7Department of Prosthodontics, New York University College of Dentistry, New York, NY 10010, USA; ki630@nyu.edu; 8Department of Operative Dentistry, Endodontology and Periodontology, Matsumoto Dental University, Shiojiri-shi 399-0704, Nagano, Japan; odu.periodontist99@gmail.com

**Keywords:** DEL-1, osteoclasts, osteoporosis, bone metabolism

## Abstract

Developmental endothelial locus-1 (DEL-1) has traditionally been characterized within the scientific community as having anti-inflammatory properties with potential inhibitory effects on osteoclast formation. Our investigation challenges this paradigm by examining *Del-1* expression in RAW264.7 cells and bone marrow-derived macrophages (BMMs) during osteoclastogenesis, as well as its functional impact on osteoclast development and activity. Our experimental findings revealed that *Del-1* mRNA levels were markedly elevated in cells stimulated by the receptor activator of the nuclear factor κB ligand compared to unstimulated precursors. When cultured with varying concentrations of recombinant DEL-1, osteoclast differentiation increased in a dose-dependent manner. Furthermore, BMMs isolated from ovariectomized mice exhibited significantly higher *Del-1* mRNA expression than those from control animals. To confirm DEL-1’s role, we employed RNA interference techniques, demonstrating that DEL-1 silencing in RAW264.7 cells substantially reduced osteoclast formation. These results suggest that DEL-1 plays a previously unrecognized role in promoting osteoclastogenesis and may contribute to bone metabolism imbalances in conditions like osteoporosis, highlighting its complex role in skeletal homeostasis and its potential as a therapeutic target.

## 1. Introduction

Bone homeostasis is maintained through a dynamic balance between bone formation and resorption, primarily mediated by osteoblasts and osteoclasts. This intricate balance is regulated by various systems, including the endocrine and immune systems, with osteoimmunological mechanisms involving macrophages and lymphocytes playing a crucial role [1,2,3,4,5,6].

Osteoporosis represents a condition characterized by altered bone turnover, resulting in net bone loss and increased fracture risk [7]. In postmenopausal women, estrogen deficiency frequently accelerates bone resorption [8]. Developmental endothelial locus-1 (DEL-1), originally identified during early mouse embryogenesis, has garnered attention for its multifaceted roles in various biological processes [9,10,11,12,13,14,15].

Previously characterized as a protein that restrains osteoclastogenesis and mitigates inflammatory bone loss, DEL-1 exhibits intricate interactions within bone metabolism mechanisms that are not fully understood. Our research aimed to provide a comprehensive analysis of DEL-1’s contribution to osteoclast development and function through experiments with murine macrophage cell lines and primary bone marrow-derived cells from an established osteoporosis model. This approach allowed us to explore DEL-1’s role in both normal physiological conditions and pathological states associated with accelerated bone resorption.

## 2. Results

### 2.1. Del-1 mRNA Expression in Osteoclasts

Our systematic analysis of DEL-1 revealed previously uncharacterized aspects of osteoclast biology. A quantitative PCR assessment demonstrated a significant upregulation of *Del-1* mRNA during osteoclast differentiation. This expression pattern was consistent across both RAW 264.7 macrophages and primary BMMs, suggesting a conserved regulatory mechanism in the osteoclast developmental pathway (Figure 1a,d). The concordance between these two distinct cell populations strengthens the biological relevance of our observation and indicates that DEL-1 may serve as an important mediator in osteoclast formation.

### 2.2. Effects of DEL-1 on Osteoclast Viability

The biological response to DEL-1 treatment exhibited nuanced concentration-dependent patterns that varied according to cellular context. Our viability assessments using both trypan blue exclusion and MTT metabolic assays revealed that at concentrations ranging from 0.1 to 100 ng/mL, DEL-1 exerted no detectable cytotoxic effects on either cell type (Figure 1b,c). Interestingly, at higher concentrations (1.0 μg/mL), we observed divergent responses between the two cell populations—RAW 264.7 cells showed reduced viability, whereas BMMs exhibited enhanced survival. This differential response highlights the cell type-specific nature of DEL-1’s effects and suggests distinct regulatory mechanisms in immortalized versus primary cells.

### 2.3. Effects of DEL-1 on Osteoclast Differentiation and Activation

Treatment with recombinant DEL-1 revealed a pronounced dose-dependent effect on osteoclast differentiation (Figure 2). Comparative analyses showed a significant increase in both the number and absorbance of osteoclasts when compared to non-treated controls. The molecular landscape of osteoclast differentiation underwent substantial transformation, with a marked upregulation of critical marker genes. Specifically, genes such as carbonic anhydrase II (*CaII*), colony-stimulating factor receptor (*C-fms*), matrix metalloproteinase-9 (*Mmp-9*), and cathepsin K (*Cat-K*) exhibited significant increases, indicating a comprehensive molecular response to DEL-1 treatment (Figure 3).

### 2.4. Effects of DEL-1 on Osteoclast Activity

Functional assessments provided deeper insights into DEL-1’s mechanisms of action. Pit assays revealed expanded resorption areas in DEL-1 treated groups, directly demonstrating enhanced bone resorption capabilities (Figure 4). These findings indicate a clear correlation between DEL-1 treatment and increased osteoclast functional potential.

### 2.5. Osteoporosis Model Investigation

The ovariectomized (OVX) mice model offered the critical in vivo validation of our experimental observations (Figure 5). Micro-computed tomography (µCT) imaging (Figure 5a) unveiled significant structural alterations characteristic of osteoporosis, including reduced trabecular bone volume, architectural disorganization, and expanded marrow cavities (Figure 5b). The molecular profiling of the OVX model confirmed our in vitro findings, showing the significantly elevated *Del-1* mRNA expression and upregulation of osteoclast differentiation markers including *Rank*, *Cat-k*, and *Mmp-9* in the bone marrow cells from OVX activated with the receptor activator of the nuclear factor κB ligand (RANKL) compared to sham-operated controls activated with RANKL (*p* < 0.01) (Figure 5c).

### 2.6. Effect of Silencing Del-1 mRNA Expression on Osteoclast Differentiation

To definitively establish DEL-1’s functional significance, we conducted small interfering RNA (siRNA) silencing experiments (Figure 6). These investigations demonstrated a remarkable reduction in DEL-1 expression, accompanied by a substantial inhibition of osteoclast differentiation when compared to random siRNA controls. Specifically, we observed a 41% decrease in DEL-1 expression and a 38% reduction in osteoclast differentiation, providing direct experimental evidence of DEL-1’s critical role in osteoclast development and function.

## 3. Discussion

Our investigation reveals novel aspects of DEL-1 biology that challenge and expand the current understanding of its role in bone homeostasis (Figure 7). While DEL-1 has been established as an important regulator in various physiological processes, its specific functions in osteoclast biology have remained inadequately characterized until now. Our findings provide compelling evidence for DEL-1’s active participation in promoting osteoclast differentiation and function, offering new perspectives on the molecular regulation of bone metabolism.

The observed upregulation of *Del-1* mRNA during osteoclastogenesis represents a paradigm shift from previous characterizations of DEL-1 as predominantly anti-osteoclastogenic. Earlier studies have portrayed DEL-1 primarily as an anti-inflammatory mediator that suppresses bone resorption. However, our data reveal a more nuanced biological role, where DEL-1 exhibits concentration-dependent effects on osteoclast development. This apparent contradiction with the established literature underscores the importance of examining protein functions within specific microenvironmental contexts and concentration ranges rather than assigning singular biological roles.

The distinct responses we observed between RAW264.7 cells and primary BMMs highlight the complexity of DEL-1’s biological interactions. This differential response pattern aligns with contemporary understanding in osteoimmunology that emphasizes the heterogeneity of cellular responses to regulatory molecules. By demonstrating that identical DEL-1 concentrations can elicit divergent outcomes in different cell populations, our study contributes valuable insights to the field’s appreciation of context-dependent signaling in bone and immune cell interactions.

Molecular marker analysis revealed profound insights into DEL-1’s mechanisms of action. The significant upregulation of osteoclast differentiation markers, including *CaII*, *C-fms*, *Mmp-9*, and *Cat-k*, provides a detailed molecular framework for understanding bone resorption processes [16,17,18]. These markers are crucial indicators of osteoclast maturation and function, suggesting that DEL-1 plays a more active role in bone metabolism than previously recognized.

The OVX mice model offered the critical in vivo validation of our experimental observation. Recent research has increasingly focused on understanding the molecular mechanisms underlying estrogen deficiency-induced bone loss [19,20,21].

The relationship between estrogen signaling and *Del-1* expression represents a particularly intriguing aspect of our findings. Estrogen is a critical regulator of bone homeostasis, primarily through its inhibitory effects on osteoclast formation and lifespan [22]. Our observation that *Del-1* expression is significantly elevated in BMMs from ovariectomized mice suggests a potential regulatory link between estrogen deficiency and DEL-1-mediated osteoclastogenesis. Several mechanisms may explain this relationship. Estrogen receptors (ERα and ERβ) function as transcription factors that regulate gene expression through binding to estrogen response elements (EREs) [23]. It is plausible that DEL-1 expression may be regulated, either directly or indirectly, through such estrogen-responsive pathways. In states of estrogen deficiency, this regulatory control may be diminished, leading to increased *Del-1* expression and enhanced osteoclastogenesis. Furthermore, estrogen deficiency is associated with the increased production of pro-inflammatory cytokines, including TNF-α, IL-1β, and IL-6, which are known to promote osteoclast differentiation and activation [24]. DEL-1 may interact with these inflammatory signaling pathways, potentially serving as a downstream mediator of estrogen deficiency-induced bone resorption. This hypothesis is supported by our observation that *Del-1* silencing significantly inhibits osteoclast formation, suggesting that targeting DEL-1 may represent a novel therapeutic approach for managing estrogen deficiency-related bone loss. Additionally, estrogen is known to influence both endothelial cell function and mesenchymal cell biology—two important cellular sources of DEL-1 [25,26]. The altered production of DEL-1 by these cell populations in estrogen-deficient states may contribute to dysregulated bone remodeling. Future studies exploring the molecular mechanisms underlying the estrogen regulation of *Del-1* expression in these cellular contexts would provide valuable insights into the pathogenesis of postmenopausal osteoporosis.

The concentration-dependent effects of DEL-1 represent a particularly intriguing aspect of our study. Modern research in molecular biology increasingly recognizes the importance of dose-dependent protein interactions [27,28]. Our observations suggest that DEL-1’s impact on osteoclast biology is far more sophisticated than a simple linear mechanism, with varying effects across different concentrations and cellular contexts.

siRNA experiments provided definitive evidence of DEL-1’s functional significance [29,30,31,32]. The substantial reduction in osteoclast differentiation following *Del-1* silencing not only confirms its critical role but also opens up potential therapeutic avenues. This approach aligns with emerging trends in targeted molecular interventions for bone metabolic disorders.

The broader implications of our research extend beyond osteoporosis. Recent studies have highlighted the interconnected nature of bone metabolism with various physiological processes, including immune response, inflammation, and systemic health [33,34,35,36]. DEL-1’s complex regulatory mechanisms suggest its potential relevance in understanding broader metabolic and inflammatory processes.

Importantly, our findings challenge the traditional view of DEL-1 as a purely anti-inflammatory protein. The concentration-dependent promotion of osteoclast differentiation reveals a more nuanced role in cellular biology. This complexity underscores the need for comprehensive, multi-dimensional approaches in studying molecular regulators of bone metabolism.

Future research directions should focus on the following aspects:Elucidating the precise molecular mechanisms underlying DEL-1’s concentration-dependent effects;Investigating potential therapeutic interventions targeting DEL-1 in bone metabolic disorders;Exploring DEL-1’s broader roles in cellular differentiation and inflammatory processes.

## 4. Materials and Methods

### 4.1. Chemicals

RANKL (R&D Systems, Minneapolis, MN, USA), *Del-1* siRNA, the siRNA transfection medium sc-36868, and the transfection reagent sc-29528 (Santa Cruz Biotechnology, Dallas, TX, USA) were used.

### 4.2. Cell Culture and Conditions

We utilized murine RAW264.7 macrophages (obtained from ATCC, Manassas, VA, USA) for in vitro experiments. These cells were maintained in Dulbecco’s Modified Eagle’s Medium (DMEM; Invitrogen Life Technologies, Carlsbad, CA, USA) supplemented with 10% heat-inactivated fetal bovine serum (FBS; Invitrogen Life Technologies) and a standard antibiotic mixture (100 U/mL penicillin and 100 μg/mL streptomycin) in a humidified incubator at 37 °C with 5% CO_2_. Prior to experimental treatments, cells were seeded and allowed to adhere for 24 h.

For osteoclast differentiation studies, RAW264.7 cells were stimulated with 25 ng/mL human recombinant RANKL (R&D Systems) and exposed to varying concentrations of mouse recombinant DEL-1 (0.1–10 ng/mL) for 4 days, with the culture medium refreshed every 48 h.

For primary cell experiments, bone marrow-derived macrophages (BMMs) were isolated from 8–12-week-old male C57BL/6J mice. After euthanasia by CO_2_ inhalation followed by cervical dislocation, femurs and tibias were aseptically removed. Bone marrow was flushed out using an α-minimum essential medium (α-MEM; Invitrogen Life Technologies) and filtered through a 70 μm cell strainer. The collected cells were centrifuged (400× *g*, 5 min), resuspended, and cultured in α-MEM containing 10% FBS and 30 ng/mL macrophage colony-stimulating factor (M-CSF; R&D Systems) for 3 days to induce macrophage differentiation. These BMMs were subsequently treated with 25 ng/mL RANKL and varying concentrations of DEL-1 under conditions identical to those used for RAW264.7 cells.

### 4.3. OVX Mice

We established an experimental osteoporosis model using four-week-old female C57BL/6J mice obtained from The Jackson Laboratory (Bar Harbor, ME, USA). Following one week of acclimatization, mice were randomly assigned to either sham-operated control or bilateral OVX groups (n = 5 per group). All procedures adhered to protocols approved by the Institutional Animal Care and Use Committee of New York University.

Sample size determination was based on power analysis using data from previous similar studies, with calculations indicating that five animals per group would provide 80% power to detect anticipated differences at α = 0.05 [13]. Prior to surgical procedures, mice were anesthetized using an intraperitoneal injection of ketamine/xylazine solution (60 mg/mL ketamine and 8 mg/mL xylazine; Phoenix Scientific, St. Joseph, MO, USA) at a dose of 1 μL/g body weight. For the OVX procedure, a small dorsal midline incision was made, the peritoneal cavity was accessed, and bilateral ovaries were identified, ligated, and excised [37]. For sham operations, the surgical procedure was identical, except that ovaries were exposed but not removed. Incisions were closed with absorbable sutures, and animals received appropriate post-operative analgesic care.

Eight weeks after surgery (allowing for the development of osteoporotic changes), mice were euthanized, and both tibias and femurs were harvested. μCT analysis was performed using a high-resolution scanner (SkyScan 1172; Bruker, Kontich, Belgium) with standardized parameters (10 μm voxel size, 50 kV, 200 μA, and 0.5 mm aluminum filter). Quantitative evaluation included the measurement of bone volume fraction (BV/TV), trabecular number (Tb.N), trabecular separation (Tb.Sp), and the structure model index (SMI), with thresholds for bone segmentation standardized across all samples.

For cellular studies, bone marrow was isolated from the harvested long bones as described in Section 4.2. BMMs from both OVX and sham-operated mice were cultured separately with M-CSF (30 ng/mL) for 3 days, followed by RANKL (25 ng/mL) stimulation, to assess *Del-1* expression and osteoclastogenic potential.

### 4.4. Real-Time RT-PCR

RAW264.7 cells and BMMs collected form C57BL/6J mice were placed in a 24-well plate (10^5^ cells/well) and added with RANKL or various concentrations (0.1–10 ng/mL) of DEL-1 at the indicated concentrations for 7 days, and total RNA was isolated from the harvested cells using a RNeasy Mini Kit (Qiagen, Valencia, CA, USA), according to the manufacturer’s instructions. The purity of total RNA was confirmed by a Nano-drop ND-1000 spectrometer (Thermo Fisher Scientific, Wilmington, DE, USA) [18,19,20,21].

Mouse Universal ProbeLibrary probes and target-specific PCR primers for *Del-1*, *C-fms*, *Mmp-9*, *CaII*, *Cat-K*, and glyceraldehyde-3-phosphate dehydrogenase (*Gapdh*), a housekeeping gene, were selected using the Probe Finder assay design software (Version 2.45; Roche) (Table 1). cDNAs were synthesized from 1 µg of total RNA for each sample using reverse transcriptase (Roche, Nutley, NJ, USA). Reactions for the 480 LightCycler (Roche) were performed in 20 µL reaction volumes for the genes encoding *Del-1*, *C-fms*, *Mmp-9*, *CaII*, *Cat-K*, and *Gapdh* using 1 µL of cDNA. *Gapdh* was used as the internal control gene to normalize the quantities of the target gene using the 2^−ΔΔCt^ method.

### 4.5. Trypan Blue Assay

We evaluated cell viability following DEL-1 treatment using the trypan blue exclusion method. RAW264.7 cells and BMMs (10^5^ cells/mL) were cultured in appropriate growth media until reaching approximately 80% confluence (typically 4 days). Cells were then incubated with DEL-1 at concentrations ranging from 0.1 to 1000 ng/mL in the presence of RANKL for 24 h. Following the treatment period, the culture medium was aspirated, and cells were gently washed three times with sterile PBS (pH 7.4). The adherent cells were then enzymatically detached using trypsin-EDTA solution (0.25%, 2 min incubation) and resuspended in a fresh medium. Equal volumes of cell suspension and 0.4% trypan blue solution were mixed and incubated for 3 min at room temperature. We determined viability by counting both stained (non-viable) and unstained (viable) cells using a hemocytometer under light microscopy, with results expressed as the percentage of viable cells.

### 4.6. MTT Assay

To quantitatively assess cellular metabolic activity following DEL-1 exposure, we employed the 3-(4,5-dimethylthiazol-2-yl)-2,5-diphenyltetrazolium bromide (MTT) colorimetric assay. RAW264.7 cells and BMMs were seeded in 96-well plates at a density of 10^5^ cells/mL in their respective growth media. After 24 h of attachment, cells were treated with a concentration gradient of DEL-1 (0.1–1000 ng/mL) in the presence of 25 ng/mL RANKL for an additional 24 h.

Following the treatment period, the culture medium was carefully aspirated to avoid cell disturbance, and 200 μL of a fresh serum-free medium was added to each well. We then added 20 μL of MTT solution (5 mg/mL in PBS; Sigma-Aldrich, St. Louis, MO, USA) and incubated the plates for 4 h at 37 °C to allow formazan crystal formation in metabolically active cells. After incubation, all of the medium was thoroughly removed, and 200 μL of dimethyl sulfoxide (DMSO) was added to each well to dissolve the formazan crystals. The plates were gently agitated on an orbital shaker for 15 min to ensure complete crystal solubilization.

Spectrophotometric readings were obtained using a Synergy HT multi-detection microplate reader (BioTek Instruments Inc., Winooski, VT, USA) at a 570 nm wavelength with a reference wavelength of 650 nm. Cell viability was calculated as the percentage of absorbance relative to untreated control wells, using the following formula: (absorbance of treated cells/absorbance of control cells) × 100. Each condition was tested in sextuplicate across three independent experiments.

### 4.7. Osteoclast Forming Assay

To evaluate osteoclast formation capacity, we performed tartrate-resistant acid phosphatase (TRAP) staining and activity assays [38]. RAW264.7 cells and BMMs were seeded in 96-well culture plates at a density of 10^4^ cells per well in a complete medium. After 24 h of attachment, cells were stimulated with RANKL (25 ng/mL) alone or in combination with varying concentrations of DEL-1 (0.1–10 ng/mL). The culture medium containing the treatments was refreshed every 48 h.

After 5 days of culture, cells were fixed with 4% paraformaldehyde for 15 min at room temperature and washed twice with PBS. TRAP staining was performed using a modified protocol based on previous studies. The staining solution contained 0.1 M acetate buffer (pH 5.0), 0.3 mg/mL Naphthol AS-MX phosphate (Sigma-Aldrich), 0.1 mg/mL Fast Red Violet LB salt (Sigma-Aldrich), and 50 mM sodium tartrate. Cells were incubated with this solution for 30 min at 37 °C in the dark.

Following staining, cells were washed with distilled water and counterstained with hematoxylin for 1 min to visualize nuclei. TRAP-positive multinucleated cells containing three or more nuclei were identified as osteoclasts and manually counted in five randomly selected fields per well under light microscopy (100× magnification). A minimum of four replicate wells were analyzed for each experimental condition, and results were expressed as the mean number of osteoclasts per well.

For the quantitative assessment of the TRAP enzymatic activity, parallel sets of identically treated cells were lysed with 100 μL of 0.1% Triton X-100 for 10 min. Lysates were then incubated with 100 μL of substrate solution containing 20 mM p-nitrophenylphosphate in 0.1 M acetate buffer (pH 5.0) with 80 mM sodium tartrate at 37 °C for 1 h. The reaction was terminated by adding 50 μL of 0.3 M NaOH, and absorbance was measured at 405 nm using a microplate reader. TRAP activity was normalized to the total protein content as determined by the Bradford assay. The numbers of TRAP-positive cells containing more than 3 nuclei were counted as osteoclasts. Representative results obtained from 4 separate wells were represented as the mean ± standard deviation (SD). Osteoclast formation was quantified using both TRAP-positive multinucleated cell counting and TRAP enzymatic activity. The number of TRAP-positive multinucleated osteoclasts (≥3 nuclei) was manually counted in five random fields per well under a light microscope [39].

### 4.8. Osteoclast Activity Assay

To assess the functional bone-resorbing activity of osteoclasts, we utilized a pit formation assay on calcium phosphate-coated plates (Bone Resorption Assay Plate 24; PG Research, Tokyo, Japan). RAW264.7 cells were seeded at a density of 1 × 10^4^ cells per well in the complete medium. Following attachment, cells were treated with RANKL (25 ng/mL) and various concentrations of DEL-1 (0.1–10 ng/mL) for 6 days, with the medium replaced every 48 h.

After the culture period, adherent cells were removed by incubation with 5% sodium hypochlorite solution for 10 min at room temperature, followed by gentle washing with distilled water three times. The plates were air-dried at room temperature for 3–4 h before analysis.

Resorption pits were visualized using a phase-contrast microscope (Nikon Eclipse TS100; Nikon Instruments, Tokyo, Japan) at 100× magnification. For each well, digital images of six randomly selected, non-overlapping fields were captured. Resorption areas were quantified using ImageJ software (Version 1.52a, NIH Image, Bethesda, MD, USA) with standardized threshold settings. The software was calibrated using a stage micrometer to ensure accurate area measurements. The total resorption area was calculated per 1.5 mm^2^ of substrate surface and expressed as a percentage of the total field area.

In parallel, we employed a fluorescence-based assessment of resorption activity. Following the removal of cells, plates were incubated with a fluorescent calcium-binding dye solution (Calcein, 5 μg/mL in PBS; Sigma-Aldrich) for 30 min in the dark. After washing, fluorescence intensity (corresponding to exposed calcium phosphate at resorption sites) was measured using a fluorescence microplate reader at excitation and emission wavelengths of 485 nm and 535 nm, respectively. Results were expressed as relative fluorescence units compared to control wells containing RANKL alone.

### 4.9. Gene Silencing Using an siRNA

*Del-1* siRNA was used to decrease *Del-1* mRNA expression and transfected using a siRNA transfection kit (Santa Cruz Biotechnology, Dallas, TX, USA), according to the manufacturer’s instructions. RAW264.7 cells (2 × 10^5^ cells/well) were placed in a 6-well plate with 2 mL antibiotic-free DMEM supplemented with 10% FBS for 24 h [14]. Additionally, 6 µL of the control and *Del-1* siRNA was diluted and duplexed in the siRNA transfection medium and reagent. The complex was mixed gently and incubated for 45 min at room temperature. The complex was placed in a plate, initially incubated for 6 h, changed with a new medium, and incubated for 4 days. These samples were immediately used for the osteoclast formation assay.

### 4.10. Statistical Analysis

All experiments were conducted with a minimum of three biological replicates per condition and repeated independently at least three times to ensure reproducibility. Data distributions were initially assessed for normality using the Shapiro–Wilk test. Given our relatively small sample sizes and the observed non-Gaussian distribution of several parameters, we adopted non-parametric statistical approaches for all comparative analyses.

For comparisons between two independent groups (such as *Del-1* expression between preosteoclasts and RANKL-activated osteoclasts), we applied the Mann–Whitney U test. For multiple group comparisons (such as dose–response experiments with different DEL-1 concentrations), the Kruskal–Wallis test was performed, followed by Dunn’s post hoc test for pairwise comparisons while controlling for family-wise error rate.

All statistical analyses were performed using GraphPad InStat software (version 3.10; GraphPad Software, La Jolla, CA, USA). Results were expressed as mean ± SD, and two-tailed *p*-values less than 0.05 were considered statistically significant. Graphical representations of data were created using GraphPad Prism software (version 8.0) with appropriate error bars and significance indicators.

Power analysis for animal experiments was conducted using G*Power software (version 3.1) based on the effect sizes observed in preliminary studies and the published literature, ensuring adequate statistical power (>0.8) for detecting anticipated differences while minimizing the number of animals used, in accordance with ethical principles.

## 5. Conclusions

Our research reveals DEL-1 as a pivotal molecule in osteoclast biology, capable of promoting differentiation and activation through concentration-dependent mechanisms. These findings challenge existing paradigms and suggest that DEL-1 may serve as a potential therapeutic target for managing bone metabolic disorders, particularly in contexts of estrogen deficiency.

## Figures and Tables

**Figure 1 ijms-26-02673-f001:**
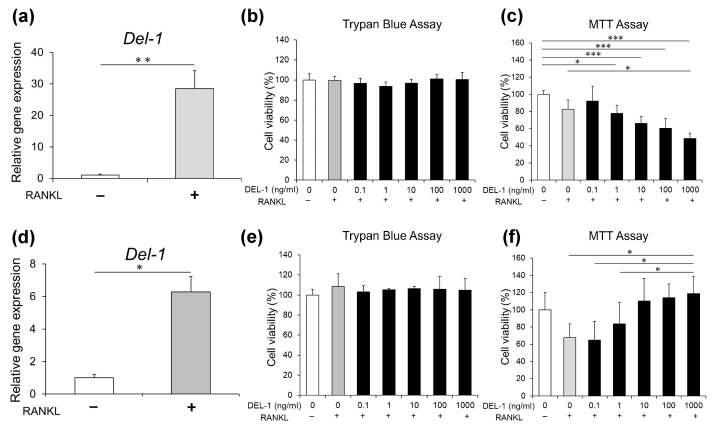
*Del-1* mRNA expression and cellular viability in RAW 264.7 cells and BMMs. After RAW 264.7 cells (**a**–**c**) and BMMs (**d**–**f**) were cultured in the presence of the receptor activator of nuclear factor κB ligand (RANKL), *Del-1* mRNA was measured and cell viability was assessed. The significant upregulation of *Del-1* mRNA expression in both RAW 264.7 cells (**a**) and BMMs (**d**) is observed in RANKL-activated cells compared to preosteoclasts. Cell viability assessment using trypan blue (**b**,**e**) and MTT assays (**c**,**f**) at various DEL-1 concentrations (0.1–1000 ng/mL). Differential cellular responses are shown between RAW264.7 cells and BMMs, highlighting concentration-dependent effects. Data are expressed as mean ± standard deviation (SD) (n = 9). * *p* < 0.05, ** *p* < 0.01, and *** *p* < 0.001.

**Figure 2 ijms-26-02673-f002:**
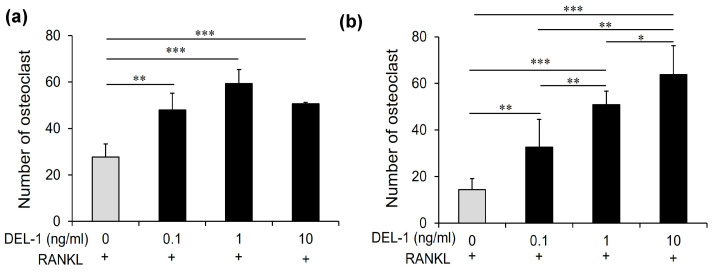
The effect of DEL-1 on osteoclast differentiation. TRAP staining images demonstrating osteoclast formation in response to DEL-1 treatment. RAW264.7 cells (**a**) and BMMs (**b**) treated with RANKL and varying concentrations of DEL-1 (0.1–10 ng/mL). The numbers of TRAP-positive cells containing more than 3 nuclei were counted as osteoclasts. Data are expressed as mean ± SD (n = 9). * *p* < 0.05, ** *p* < 0.01, and *** *p* < 0.001.

**Figure 3 ijms-26-02673-f003:**
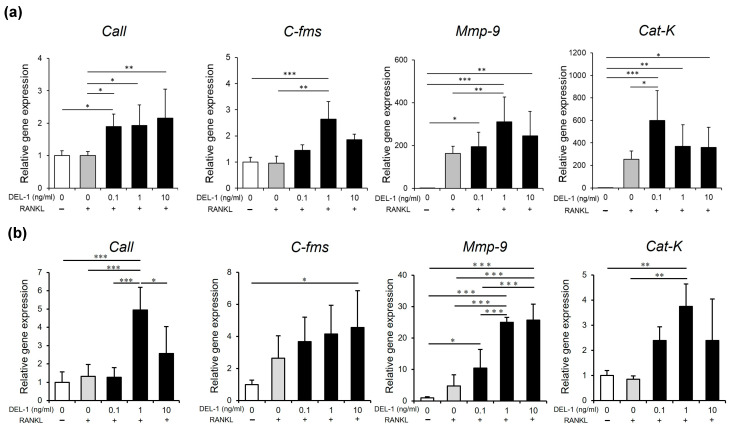
Osteoclast differentiation marker gene expression. Real-time RT-PCR analysis of osteoclast differentiation marker genes in RAW264.7 cells (**a**) and BMMs (**b**). Expression levels of carbonic anhydrase II (*CaII*), colony-stimulating factor receptor (*C-fms*), matrix metalloproteinase-9 (*Mmp-9*), and cathepsin K (*Cat-K*) in RAW264.7 cells and BMMs under various DEL-1 concentrations. Significant upregulation of differentiation markers is observed, indicating comprehensive molecular response to DEL-1 treatment. Data are expressed as mean ± SD (n = 9). * *p* < 0.05, ** *p* < 0.01, and *** *p* < 0.001.

**Figure 4 ijms-26-02673-f004:**
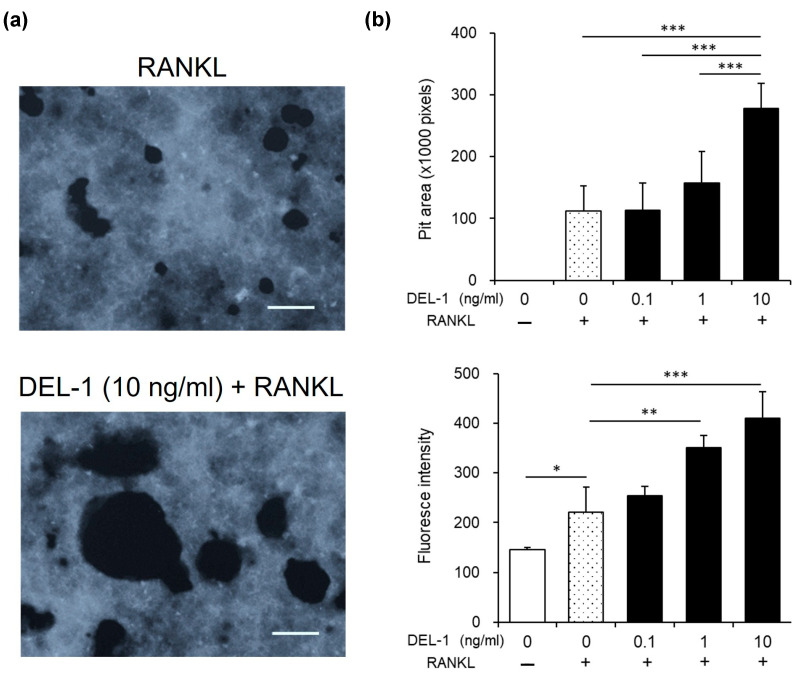
The effect of DEL-1 on osteoclast activation. Pit resorption assay demonstrating enhanced bone resorption capabilities. Pit area images showing expanded resorption areas in DEL-1 treated groups (**a**). Quantification of pit areas using ImageJ (Version 1.52) (**b**). Comparative analysis demonstrates the functional implications of DEL-1 treatment on osteoclast bone resorption capabilities. Original magnification × 10, bars = 100 μm. Data are expressed as mean ± SD (n = 6). * *p* < 0.05, ** *p* < 0.01, and *** *p* < 0.001.

**Figure 5 ijms-26-02673-f005:**
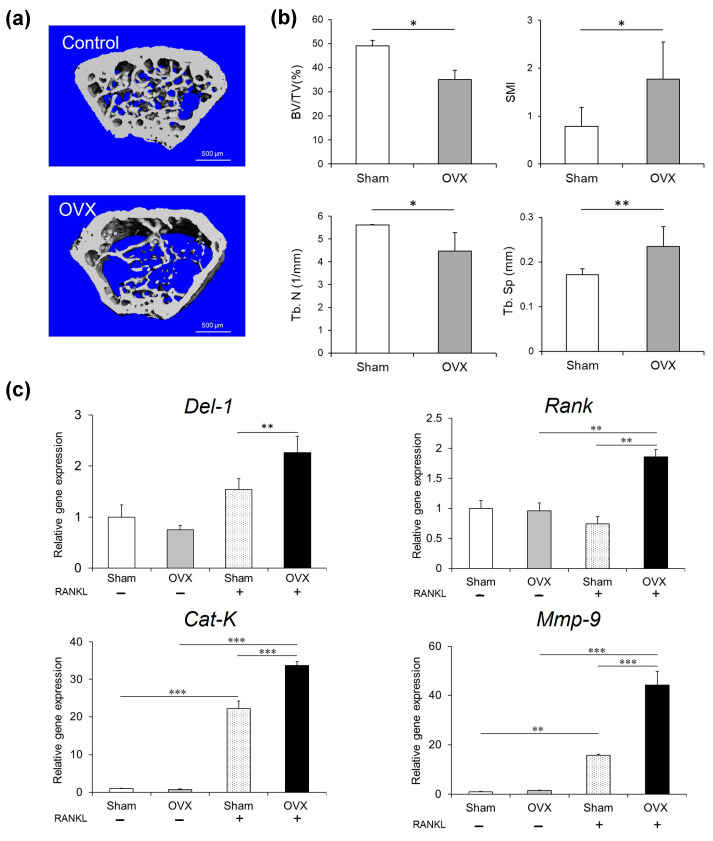
Ovariectomized (OVX) mouse model. Micro-computed tomography (µCT) imaging of bone metabolism changes. Representative µCT images showing structural alterations in OVX mice compared to sham-operated controls (Control) (**a**). Bars = 500 μm. Quantitative analysis of bone volume fractions, trabecular separation, and structure model index (**b**). Abbreviations indicate bone volume (BV), total volume (TV), structure model index (SMI), trabecular separation (Tb. Sp), and trabecular number (Tb.N). The mRNA expression of *Del-1*, *Rank*, *Cat-K*, and *Mmp-9* was measured in BMMs collected from mice with OVX and/or RANKL by real-time RT-PCR (**c**). Molecular profiling of BMMs demonstrating elevated *Del-1* mRNA expression and upregulation of osteoclast differentiation markers in OVX mice. Data are expressed as mean ± SD (n = 5). * *p* < 0.05, ** *p* < 0.01, and *** *p* < 0.001.

**Figure 6 ijms-26-02673-f006:**
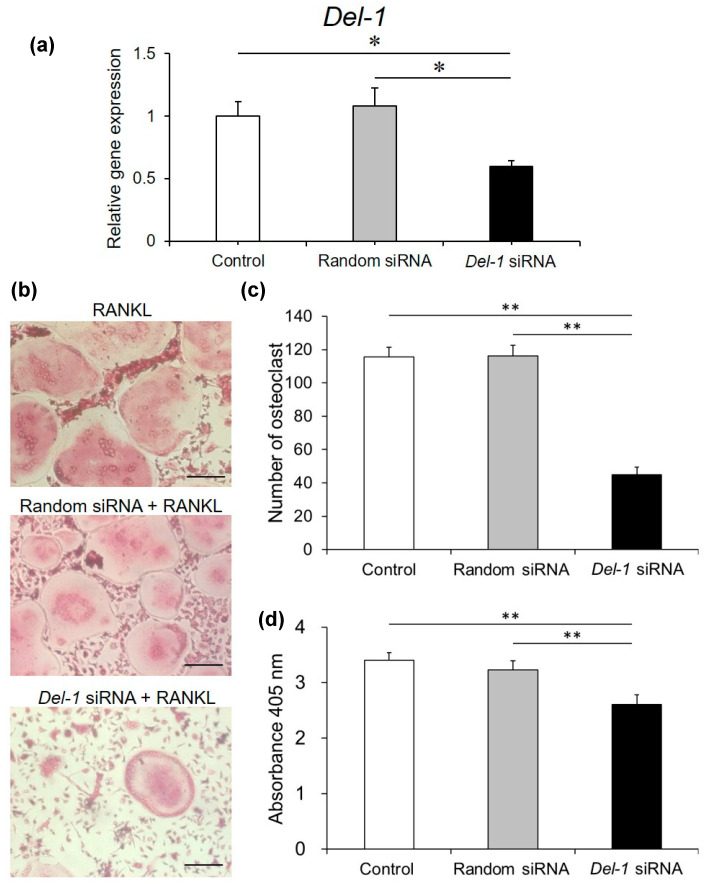
*Del-1* siRNA suppressed osteoclast differentiation. Small interfering RNA (siRNA) experiments revealed the functional significance of DEL-1 in RAW264.7 cells. Confirmation of *Del-1* mRNA expression reduction using siRNA (**a**). TRAP staining images showing the effect of *Del-1* siRNA on osteoclast differentiation (**b**). Original magnification × 10, bars = 100 μm. The numbers of TRAP-positive cells containing more than 3 nuclei were counted as osteoclasts (**c**). Quantification of TRAP-positive multinucleated cells demonstrating 38% reduction in osteoclast differentiation. Osteoclast formation was measured by a TRAP-solution assay (**d**). Statistical analysis of *Del-1* mRNA expression and osteoclast differentiation following siRNA treatment (**d**). Data are expressed as mean ± SD (n = 9). * *p* < 0.05 and ** *p* < 0.01.

**Figure 7 ijms-26-02673-f007:**
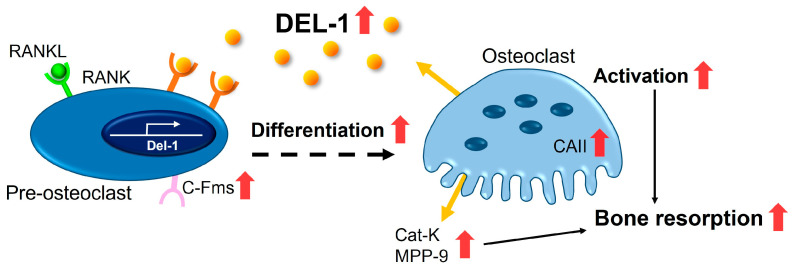
Proposed mechanism of bone resorption regulation by DEL-1. This schematic diagram illustrates the molecular pathways through which DEL-1 promotes osteoclast differentiation and activation. The left panel depicts how DEL-1 interacts on osteoclast precursors, activating intracellular signaling cascades via RANK-RANKL interaction. These signals upregulate the expression of osteoclast-specific genes (*Cat-K*, *MMP-9*, etc.) and promote the cytoskeletal reorganization necessary for mature osteoclast formation. The right panel illustrates how estrogen deficiency alters this pathway, showing increased *Del-1* expression and enhanced RANKL signaling, collectively promoting excessive osteoclastogenesis and bone resorption. Red arrows indicate upregulation.

**Table 1 ijms-26-02673-t001:** Primer sequences for real-time RT-PCR.

*Del-1*	Forward	ACAGGCATCATTACACAAGGAG
	Reverse	GGGAGGGTCGATGACATTTTT
*C-fms*	Forward	GGTTGTAGAGCCGGGTGAAA
	Reverse	AAGAGTGGGCCGGATCTTTG
*Mmp-9*	Forward	GGGTCTAGGCCCAGAGGTAA
	Reverse	TAACGCCCAGTAGAGAGCCT
*CaII*	Forward	GACCCAGGTGTCTCATGTGG
	Reverse	GACGCCAGTTGTCCACCATC
*Cat-k*	Forward	CTCAACAGCAGGATGTGGGT
	Reverse	TTCAGGGCTTTCTCGTTCCC
*Gapdh*	Forward	AACGACCCCTTCATTGAC
	Reverse	TCCACGACATACTCAGCAC

## Data Availability

Data will be made available on request.

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
