# Peer review of "Developmental Endothelial Locus-1 Promotes Osteoclast Differentiation and Activation"

_ijms, 2025, doi:10.3390/ijms26062673_

Round 1
Reviewer 1 Report (New Reviewer)
Comments and Suggestions for Authors
This study was aimed to focus on Developmental Endothelial Locus-1 (DEL-1) and its role in osteoclast differentiation and osteoporosis, which is a relevant topic in bone metabolism and osteoimmunology. The current proposal is interesting and well-written. Therefore, I recommend that the current study be published after major revisions as follows:
1- Please provide scoring for figures:
- Figure 4a (Pit assay) requires scoring details to better interpret DEL-1’s effect on osteoclast activity.
- Figure 5a (Micro-CT images of OVX model) should include a scoring system or quantification to support structural changes.
- Figure 6b (siRNA experiment) needs an objective scoring method to quantify osteoclast differentiation.
2- Could the authors refer to Estrogen and DEL-1 Relationship:
o The authors should discuss the link between estrogen deficiency and DEL-1 expression more explicitly.
o How does estrogen influence DEL-1-mediated osteoclastogenesis? Does DEL-1 serve as a mediator of estrogen deficiency-induced bone resorption?
3- The manuscript lacks an in-depth discussion on the molecular mechanisms by which DEL-1 modulates osteoclastogenesis. Please provide a diagrammatic figure illustrating DEL-1’s signaling pathways in osteoclast differentiation.
4- Many techniques (OVX model, TRAP staining, RT-PCR, siRNA transfection) lack appropriate references to previous studies. Proper citations should be included.
5- Statistical comparisons require references for methodology. Were non-parametric or parametric tests chosen based on normality tests? Justification is needed.
Author Response
Please see the attachment.

Reviewer 2 Report (New Reviewer)
Comments and Suggestions for Authors
Abstract
Lines 18-19: I think authors should be more cautious regarding the statement “(DEL-1) has been recognized as a potent anti-inflammatory protein with potential anti-osteoclastogenic effects”. They should include a more conservative and general view regarding the role of DEL-1 (because the authors’ results showed the opposite).
Line 20: Please specify the cell types used.
Line 24: Regarding “Del-1 mRNA expression was notably higher in ovariectomized mice models” – please detail in which this was performed. It was in the BMMs collected from ovariectomized mice? This is linked to the results presented in Fig. 5.
Line 25: Please specify the cell type used.
Introduction
Lines 43-44: Regarding the sentence “…. examine DEL-1’s role in osteoclast differentiation and activation using murine osteoclasts and preclinical osteoporosis models”. The last part, regarding preclinical osteoporosis models, it should be mentioned that studies were conducted with BMMs collected from the animals (studies were not conducted in the animals).
Results
Fig. 5 – mRNA analysis were conducted in which cells? Please specify in the legend of the Figure.
Fig. 6 – Please mention the cell type.
Methodology
I think the entire section lacks detailed information. It is not clear to the reader – it does not allow to replicate the experiments. Some examples:
Section 4.2. How many days RAW cells and BMMs were cultured before the addition of DEL-1. Cultures were exposed to RANKL+DEL-1 for 4 days? Details on the BMMs used are relevant: origin/type of mice, isolation procedure, culture conditions before (without or with RANKL?) and after adding RANKL+DEL-1.
All the culture conditions were similar for RAW cells and BMMs? RAW are already macrophages, but BMMs are earlier precursors (normally, they need previous treatment with M-CSF to get the macrophages).
Section 4.3. How long were rats maintained after ovaries had been removed and before being euthanized to collect the bone marrow? Please specify the culture conditions for bone marrow cells (before and after adding DEL-1).
What is the difference between the Bone marrow cells mentioned in section 4.2 and the bone marrow cells from the sham animals in section 4.3? They were from the same type of mice?
Section 4.4. It is described that cells were exposed for 3 and 7 days. Which day was used to conduct the PCR analysis? Apparently, in the Figures, data refer to only one culture time. And, regarding BMMs cells – it refers to which BMMs? (cells described in section 4.2 or cells referred in section 4.3?
Section 4.6. Culture conditions before adding DEL-1 are missing.
Section 4.8. (Lines 303-304). It is described that osteoclast activity was measured using fluorescence with … How this assay was done? Please explain and clarify.
Round 2
Reviewer 2 Report (New Reviewer)
Comments and Suggestions for Authors
The authors greatly improved the manuscript.
This manuscript is a resubmission of an earlier submission. The following is a list of the peer review reports and author responses from that submission.
Round 1
Reviewer 1 Report
Comments and Suggestions for Authors
This manuscript has demonstrated that low concentrations of DEL-1 promote osteoclast differentiation of RAW264.7 cells, contrary to previous reports. However, only one cell line was used, which is insufficient to validate the effect of DEL-1 on osteoclast differentiation. To clarify that these phenomena are not specific to the cells the authors obtained, other cell lines or BMMs should be used to demonstrate that this is general phenomenon in osteoclast differentiation.
1. The authors should clearly describe at what density, on which plates, and for how many days the cells were cultured in each experiment. Some descriptions in figure legends and Materials and Methods are inconsistent.
2. Although the authors described that higher concentrations of DEL-1 have a cytotoxicity effect, their MTT assay measures mitochondrial reductase activity, so it is not suitable for measuring the cell viability of fusing cells such as osteoclasts.
3. The margin of error in qPCR in cells added with recombinant DEL-1 is so large that RNA purification appears to be unsuccessful.
4. The absorbance in Figure 5b exceeds 3, it is recommended to check whether it is within the measurable range of the absorbance meter.
5. The authors are trying to clarify the relationship between DEL-1 and bone resorption diseases using OVX mice and periodontal disease mice. However, the elevation of Del-1 expression didn’t differ from that of other osteoclast markers. It seems that these model mice only increased the number of osteoclasts rather than up-regulating Del-1 expression. It is necessary to devise methods such as inhibiting Del-1 in these model animals.
6. siRNA experiments should show suppression of Del-1 mRNA or DEL-1 protein. It should be also be checked how the expression of osteoclast differentiation marker genes is affected.
Reviewer 2 Report
Comments and Suggestions for Authors
The research presented in this manuscript is very interesting. The suggestion is to explain how the sample size was determinated and to cite more recent literature, given that 60% of the references are older than 10 years.
Reviewer 3 Report
Comments and Suggestions for Authors
General Comments
The authors examined the expression and function of Del-1 in osteoclasts and found that its function is contrary to what has been reported in the past. Although this is an intriguing finding, the manuscript was written insufficiently and needs more data to verify the author’s conclusion. The following refinements should be needed.
Major Comments
I. How many samples per group were used in each experiment? This information must be needed to assure the quality of the results in this study.
II. Although the authors state that cell viability was decreased with dose-dependent Del-1 addition (Figure 1b), the significant differences were only shown in comparison with preosteoclasts, which were cultured without RANKL. If there is no significant difference compared to osteoclasts, which were cultured without Del-1, the result must be interrupted as there was no change in the cell viability. Please clarify how recombinant Del-1 affects cell viability in osteoclasts, probably using some higher concentrations of Del-1 such as 0.5, 1.0, or 2.0 µg/ml.
III. Regarding real-time RT-PCR, how many samples per group were analyzed? In addition, have the authors performed technical and biological replicates? The method section should be written more thoroughly to verify that the authors performed experiments properly.
IV. Which time point was used for real-time RT-PCR? In the method section, it was written that total RNA was harvested after 3 or 7 days of culture, but it was shown only one-time point in Figure 2d and was not clear since there was no description in the figure caption.
V. The ligature-induced periodontitis model should not be included in this manuscript since the data was not shown enough. Moreover, the Del-1 expression in Figure 4e was not from osteoclast. It was probably from gingival tissue based on the method/discussion section, but not clear. I suggest that this manuscript should focus on Del-1 expression in osteoclast or cell sources in bone tissue. If authors discuss Del-1 expression in various cell types and bone loss in the periodontitis model, they need to perform more experiments focused on which kind of cell sources provide Del-1, how Del-1 expressions affect ligature-induced periodontitis in mice and discuss the difference of Del-1 function compared to the past studies, especially Shin et al.
VI. How was the inhibition rate of Del-1 expression by siRNA treatment? Show the relative Del-1 mRNA expression determined by real-time PCR in RAW264.7 cells transfected (or not) with Random siRNA, or Del-1 siRNA.
VII. The authors addressed there was no effect on osteoblast but didn’t show the results (page 7, lines 206-207). I assume this is critical to conclude the Del-1 effect on bone metabolism. To support the author’s conclusion in the OVX model, cell viability, osteoblast differentiation, and its function should be examined and addressed using MC3T3-E1 cells and recombinant Del-1. Also, the immunohistochemistry of Del-1 in bone tissue in the OVX model would strongly corroborate the author's conclusions.
Minor Comments
a. All histological images need a scale bar i.e., in Figures 2a, 4d, and 5a. Please burn the scale and indicate how long it was in the figure caption.
b. Unnecessary sentences, such as the description in the format, remains (page 6, lines 135-137).
c. There seems to be no description of which error bars were used. It must be indicated which standard deviation (SD) or standard error of the mean (SEM) was used.
d. It seems that the concentrations in the Figure 1b legend were between 0.1 – 100 ng/ml, not 0.1 – 10 ng/ml (page 2, line 70).
e. Acknowledgement must be prepared correctly.
Round 2
Reviewer 1 Report
Comments and Suggestions for Authors
All comments should be responded to whether the authors agree with them or not.
I suggested experimenting with other cell lines or BMMs, but the authors did not. Since the authors used BMMs in Fig. 4c, it should not be difficult to study the effects of DEL1 on BMMs. The effect of DEL-1 shown by the authors inconsistent with that of Ref.14, which was investigated using several osteoclast differentiation models. The experiments done only with RAW264.7 cells look less convincing.
1. The methods have been improved.
2. Again, the MTT assay measures mitochondrial reductase activity. The activity of mitochondrial enzymes in RANKL-untreated macrophages and fused osteoclasts should not be comparable, and comparisons between the two do not reflect cell viability. In the previous manuscript, comparisons between RANKL-treated cells were not made, so I commented as such.
3. The margin of error in qPCR in cells added with recombinant DEL-1 is so large. Since there is no problem with RNA purification, the timing of RNA collection should be reconsidered.
4. Since there is a difference of more than 100 times in absorbance between Fig. 2c and Fig. 5d, I thought that the absorbance was written incorrectly, but I believe that the absorbance is within the measurable range of the meter. However, absorbance above 1.0 is outside the valid range of Beer-Lambert’s Law. Moreover, the results in Fig. 2c and Fig. 5d differed by more than 100-fold despite comparable cell concentrations, compromising the reliability of this assay.
5. Again, the experiments in OVX mice have not examined the effects of DEL-1. It is necessary to devise methods.
6. It should be checked how the expression of osteoclast differentiation marker genes is affected. Since the effect of Del1 inhibition in RAW264.7 cells is opposite to Ref. 14, it is very important to confirm the expression of osteoclast differentiation marker genes.
Reviewer 3 Report
Comments and Suggestions for Authors
The authors updated the manuscript according to the reviewers' comments, however, there still needs some improvements. The followings would be needed to support the author’s conclusion.
1. The updated Figure 1b shows significant differences only between 1000 ng/ml DEL-1 compared to 0 ng/ml DEL-1 under the existence of RANKL, which means there was no difference observed regarding cell viability between 0.1 to 100 ng/ml addition of DEL-1. Is this correct? If so, the sentences in abstract and the line 64 on page 2 don’t make sense. Please carefully double-check the results and address the results properly.
2. Why authors didn’t include the result of 4.0 µg/ml DEL-1 addition on osteoclast although they performed? It was addressed in the response letter but wasn’t in any section of the manuscript.
3. The updated Figure 2 shows 4 panels of the results, but all results were addressed as “a)”. There should be “a)” to “d)”.
4. Figure 4 still shows the results of the periodontitis model (4d and 4e).
5. Please revise the sentence in lines 245 and 246. Since the authors deleted the periodontitis model, there were no results related to “inflamed tissues”.
6. I still believe the results of osteoblast would be needed for this manuscript. Since authors have already performed it, the results should be included not only in sentences but also in figures, in the M&Ms, and in the Results section. Verifying there was no effect on osteoblast by DEL-1 will support the author’s conclusion strongly.